# Fetal Alcohol Spectrum Disorders and Inadequacy of Care: Importance of Raising Awareness in Clinical Practice

**DOI:** 10.3390/children11010005

**Published:** 2023-12-20

**Authors:** Sigita Lesinskienė, Emilijus Žilinskas, Algirdas Utkus, Rūta Marčiukaitytė, Gabrielė Vasiliauskaitė, Rugilė Stankevičiūtė, Odeta Kinčinienė

**Affiliations:** 1Clinic of Psychiatry, Institute of Clinical Medicine, Faculty of Medicine, Vilnius University, 03101 Vilnius, Lithuania; 2Faculty of Medicine, Vilnius University, 03101 Vilnius, Lithuania; emilijus.zilinskas@mf.stud.vu.lt (E.Ž.); ruta.marciukaityte@mf.stud.vu.lt (R.M.); gabriele.vasiliauskaite@mf.stud.vu.lt (G.V.); rugile.stankeviciute@mf.stud.vu.lt (R.S.); 3Department of Human and Medical Genetics, Institute of Biomedical Sciences, Faculty of Medicine, Vilnius University, 03101 Vilnius, Lithuania; algirdas.utkus@mf.vu.lt; 4Clinic of Children’s Diseases, Institute of Clinical Medicine, Faculty of Medicine, Vilnius University, 03101 Vilnius, Lithuania; odeta.kinciniene@mf.vu.lt

**Keywords:** fetal alcohol syndrome, fetal alcohol spectrum disorders, prenatal alcohol exposure

## Abstract

Prenatal alcohol exposure is one of the major avoidable causes of developmental disruption and health abnormalities in children. Fetal alcohol spectrum disorders (FASDs), a significant consequence of prenatal alcohol exposure, have gained more attention recently. This review aims to provide a narrative approach to the scientific literature on the history, clinical presentation, diagnosis, and management of FASDs. A literature search in PubMed, ScienceDirect, and Google Scholar online databases was conducted. The dates of publications ranged from 2000 to 2023. FASD presentations tend to persist into adulthood, which, combined with environmental factors, potentially lead to secondary psychosocial problems and disabilities. This review covers different aspects of FASDs regarding the concept of the umbrella term and public health, somatic, and psychiatric perspectives. FASD management remains an obstacle to health professionals, and mental health problems are underestimated. Its management involves a multi-disciplinary team, which varies according to the patient’s individual needs. FASD diagnosis and management have not been sufficiently established and tailored. Stigma, cultural contexts, knowledge gaps, and the heterogeneity of clinical manifestations are significant barriers to an accurate diagnostic process. Further development of early interventions and the elaboration of complex treatment approaches are needed.

## 1. Introduction

It is difficult for specialists working in clinical practices in Lithuania to diagnose fetal alcohol spectrum disorders (FASDs), especially in milder cases in older children. Clinicians may wish to protect a child with an FASD from stigmatizing diagnostic labels in the healthcare system, mistakenly believing that not naming the causes of the disorder will facilitate their psychological state and social adaptation. When discussing these issues with colleagues from other countries, it became clear that these topics are relevant and incompletely resolved. Research shows the prevalence of FASDs in 76 countries is > 1% and is high in individuals living in out-of-home care or engaged in justice and mental health systems [1]. Diagnosis of milder cases in children, especially in those who have grown up in families, could be missed due to stigma, and they do not receive comprehensive assessments, maltreatment prevention, relevant treatment, and multisectoral management of the complex help they need. Children with these disorders not only need thorough examinations but also appropriate therapeutic communication explaining their FASD-related difficulties and the nature of the disorders. Talking about these topics is not easy for professionals, neither with children nor their parents or guardians. The organization of help is not clearly defined, and most articles indicate the need for further research on the issues of FASD identification and the organization of long-term help and complex treatment. We examined the academic literature and determined that many questions remain unanswered in the broad FASD field [2,3]. The social and cultural vulnerability of individuals with FASDs, stigma, quality of life, behavioral and learning problems, long-term services, and cooperation among help providers remain unsolved clinical and scientific problems.

Alcohol consumption during pregnancy is a common public health issue [1,4]. Fetal alcohol spectrum disorders, a burdensome consequence of prenatal alcohol exposure, have gained more attention in the recent scientific literature [1,5,6]. The research shows that supportive alcohol policy could serve as a key element of FASD prevention and the promotion of women’s and fetal health [7]. A considerable part of the research regarding FASDs focuses on its pathogenesis, neurological findings, public awareness, stigma, obstacles to complex diagnostic measures, prevention strategies, and follow-up care [2,8,9]. However, the latter aspects are most often discussed separately. 

Thus, this article aims to provide a coherent narrative review summarizing recent and critical aspects of FASDs from clinical and public health perspectives. The occurrence of FASDs at all ages is discussed and presented visually in diagrams, highlighting the most critical and relevant issues in recognizing and helping children and adolescents with FASDs. Reducing stigma and finding appropriate ways to talk to mothers and their children about this sensitive topic are also explored.

## 2. Materials and Methods

An extensive non-systematic narrative literature review was performed. Articles published from 2000 to 2023 were included in the review. The search was conducted in the PubMed, ScienceDirect, and Google Scholar online databases. Specific keywords—”fetal alcohol syndrome”, “FAS”, “fetal alcohol spectrum disorders”, “FASD”—were used along with the terms “diagnostic criteria”, “epidemiology”, “burden”, “literacy”, “stigma”, “follow-up”, “care”, and “prevention”. Only peer-reviewed publications written in English were reviewed.

## 3. Recent Findings

### 3.1. The Concept of FASD

Fetal alcohol spectrum disorders are an umbrella term used to describe a pattern of disabilities and abnormalities that result from fetal exposure to ethanol during pregnancy and are the most common non-heritable causes of intellectual disability. The effects on the fetus may include physical, mental, behavioral, and/or learning disabilities, with possible lifelong implications, and encompass a phenotypic range that can greatly vary between individuals but reliably include one or more of the following: facial dysmorphism, fetal growth deficiency, central nervous system dysfunction, and neurobehavioral impairment. Five diagnoses fall under the term FASDs: FAS (fetal alcohol syndrome); partial FAS (partial fetal alcohol syndrome); alcohol-related neurodevelopmental disorder (ARND); neurobehavioral disorder associated with prenatal alcohol exposure (ND-PAE); and alcohol-related birth defects (ARBDs) [10]. This was the first systematic approach delineating the diagnostic categories introduced by the Institute of Medicine (IOM), which was revised in 2005 and 2016 and led to FASDs being divided into five subtypes. The fetal alcohol spectrum disorders’ four-letter diagnostic code was developed in 2000 to give greater diagnostic scope for describing children adversely affected by alcohol who did not fulfill the diagnostic criteria for FAS. One study assessed several diagnostic frameworks, wherein authors noted a lack of ascribed convergent validity, including the absence of a diagnostic gold standard against which to measure competing systems. Such a problem occurs when there is a debate between diagnoses. For instance, according to one diagnostic framework, a patient can be diagnosed with FAS and, according to another, receive no diagnosis. The authors noted that the concordance between systems is improved when diagnostic categories collapse into FASDs versus no diagnosis. Such an approach was designed for the 2016 Canadian system. The Canadian system introduces two subtypes of criteria: FASDs with sentinel facial features and FASDs without sentinel facial features [5]. Physical symptoms, which can be identified phenotypically, are characteristic facial features, such as short palpebral fissures, a thin vermilion border of the upper lip, and a smooth philtrum. Growth retardation is defined as being in the 10th percentile or less using height and weight measurements on standard growth curves adjusted for age, sex, race, or ethnicity [10]. Externally visible symptoms make it possible to notice possible disorders earlier, especially in early childhood; therefore, FAS and pFAS can be more recognizable compared with Alcohol-Related Neurodevelopmental Disorder (ARND), Neurobehavioral Disorder Associated with Prenatal Alcohol Exposure (ND-PAE), and Alcohol-Related Birth Defects (ARBD) because of the characteristic facial dysmorphology and growth retardation [11]. The diagnostic criteria for the conditions of FASDs are presented in Table 1.

### 3.2. Epidemiology 

Despite many public education programs regarding alcohol consumption during pregnancy, the percentage of pregnant women consuming alcohol is increasing. According to data from the Centers for Disease Control and Prevention, alcohol consumption in pregnant women increased from 7.6% in 2012 to 10.2% in 2015 [10]. Furthermore, the number of pregnant women reporting binge drinking (four or more alcoholic beverages at once) increased from 1.4% to 3.1% [10]. In addition, data from the 2009 National Birth Defects Prevention Study of over 4000 women indicate that 30% of pregnant women use alcohol and 8% engage in binge drinking on at least one occasion [4].

High rates of alcohol consumption during pregnancy allow us to suspect a high prevalence of fetal alcohol spectrum disorders. The global prevalence of FASDs in children and youth in the general population is 7.7 per 1000 people. The highest prevalence ratio is in the European Region (19.8 per 1000 people) [14]. One study in the US indicated that 1.1 to 5% (depending on the community) of first-graders have FASDs, and the weighted prevalence estimates for FASDs in this population ranged from 31.1 to 98.5 per 1000 children [15]. 

### 3.3. Burden of FASDs

FASDs have an impact on various individual and social aspects. The outcomes caused by FASDs can be divided into primary and secondary disabilities. Primary disabilities comprise congenital conditions caused by the teratogenic effects of alcohol on the fetus’s brain and differences in behavior and cognition [16]. Delayed growth and craniofacial anomalies, including palpebral fissures, a smooth philtrum, and a thin upper lip vermilion, though infrequent, can help in terms of differential diagnoses [17]. In addition, FASDs have an increased risk of comorbidities, with ADHD, depression, anxiety disorder, oppositional defiant disorder, and receptive as well as expressive language disorders being commonly reported [18]. FASDs have been associated with adverse childhood experiences (ACEs), which begin with prenatal alcohol exposure and tend to continue across the lifespan. In one systematic review, it was estimated that ADHD had the highest prevalence of 50% among individuals diagnosed with FASDs [19]. The impairments described above as well as adverse childhood experiences (ACEs) can increase the risk of comorbidities. FASDs and subsequent exposure to ACEs, with neglect, parental substance abuse, parental separation or divorce, and physical abuse being the most prevalent, can lead to the development of additional neurobehavioral disorders. Moreover, children with FASDs living in foster care exhibited higher ACE scores than non-FASD subjects (RR = 9.05). Children diagnosed with FASDs were 9 times more likely to be placed in foster care and 6.7 times more likely to be placed in residential care compared with controls [20,21]. Secondary disabilities may include poorer school performance, unemployment, needing support from social welfare, an increase in psychiatric and medical comorbidities, problems with the law, and victimization [16,22]. Problems with the justice system, victimization, and incarceration may also be relevant to FASDs [22]. Disruptions in adaptive functioning, including poor communication and social and daily living skills, as well as an inability to properly regulate behavior and emotions, can lead to daily challenges at school, home, and community environments, increasing the risk of disrupted school experiences, alcohol or drug addictions, inappropriate sexual behavior, and a reduced capacity to live independently [16]. Primary and secondary disabilities make significant demands of foster parents, birth parents, schools, mental health care systems, and legal institutions [22]. The biopsychosocial burden of FASDs is summarized in Figure 1.

### 3.4. Assessing Prenatal Alcohol Exposure

Documented prenatal alcohol exposure is a core component in making an FASD diagnosis. To effectively address alcohol consumption during pregnancy, an appropriate assessment of alcohol use as a routine component of prenatal care is essential. Many alcohol assessment tools in reproductive-age women and during pregnancy have been proposed (for example, CAGE, TWEAK, AUDIT-C, T-ACE, and T-ACER3) [23]. The biggest concern regarding such tools is their reliance on direct questioning and self-reporting, thus being prone to under-detection due to women’s unwillingness to admit to acts of alcohol consumption. Women who use substances during their pregnancies feel fear and stigma and, thus, are less willing to cooperate with physicians and more likely to avoid treatment [24]. The situation can be even more complicated in cases of maternal polysubstance abuse. There are efforts to create biomarkers that may be more valid than maternal self-reports. For example, one systematic review found that the detection of fatty acid ethyl esters in meconium indicated a more than four times higher prevalence of prenatal alcohol exposure compared with maternal self-reporting [25]. The latter findings indicate that maternal self-reports regarding PAE may not be sufficient as a sole information source [3,7].

### 3.5. Recognition of Signs over Time

Children with dysmorphic features of FASDs can be identified as early as 9 months of age, with a peak of identification at 18 months of age compared with an unaffected group of children. However, children with severe impairment of dysmorphic features can be identified at birth. The literature notes that although early behavioral and development impairment is usually noted between 18 and 42 months of age, CNS impairment may not be apparent until the child is in school [26]. Thus, early childhood assessment appears the most sensitive, as well as providing sometimes the only evidence of FASD phenotypes. Children with FASD phenotypes can be missed if they do not have an assessment of the disorder in time, primarily due to the diminishing prevalence of short palpebral features and a small head circumference, as well as the physical features of prenatal alcohol exposure being absent in as many as 75% of affected children [27]. This statement is supported by a US study highlighting that without early childhood screening, only one in seven children will be identified later in their lifetime. Therefore, this confirms the importance of early screening before children start pre- or primary school as early interventions are the most effective in improving children’s future quality of life [28].

### 3.6. Challenges in Follow-Up Care

Neurocognitive habilitation therapy and social skills improvement programs are especially recommended for patients with an FASD diagnosis. An effective neurocognitive habilitation program is “Alert” [29]. During the program, patients are encouraged to work in groups and introduced to learning concepts, creative activities, and the acknowledgment of sensory functions. Therefore, children learn to adapt to the community. If needed, medications can be administered (e.g., in the case of ADHD, depression, and anxiety disorders). Parents should also be educated on communicating with their children.

Other interventions that aim to help patients to adapt may also be effective. For example, an analysis of one intervention, “MILE” (The Math Interactive Learning Experience), showed a significant gain in the patient’s math skills and behavior; another intervention, called “CFT” (Child Friendship Training), improves social skills, whereas “CPAT” (Computerized Progressive Attention Training) decreases reaction times and improves attention spans [30]. Cognitive control therapy significantly changes a child’s behavior. It teaches self-regulation, self-observation, and different thinking.

The management of other comorbidities is chosen by a multidisciplinary team depending on the specific patient. Usually, a multidisciplinary team combines an audiologist, cardiologist, developmental pediatrician, developmental therapies, family therapist, nephrologist, neurologist, occupational therapist, ophthalmologist, physical therapist, primary care physician, psychiatrist, psychotherapist, sensory integration therapist, social worker, special education teachers, and speech-language pathologist. The literature shows that the lack of a functioning multidisciplinary team arises from the absence of FASD follow-up guidelines and neuropsychological help, and patients are left without proper care [2,3,10]. 

A summary of the challenges in evaluating prenatal alcohol exposure, diagnosing FASDs, and providing adequate care afterward is presented in Figure 2.

## 4. Discussion

This review presented different aspects of FASDs regarding the concept of the umbrella term and public health, somatic, and psychiatric perspectives. Despite the significant prevalence of FASDs, there is a lack of knowledge about prenatal alcohol consumption and its consequences. Based on data from the Eurobarometer Report in 2010, 8% of European citizens tend to disagree or totally disagree that alcohol consumption increases the risk of birth defects and 8% have doubts [31]. Women’s knowledge regarding alcohol consumption during pregnancy is also incomplete. According to one study from Australia by E. Peadon et al., only 61.5% of pregnant women had heard about the effects of alcohol on a fetus, and 55.3% had heard about fetal alcohol syndrome [32]. Furthermore, FASD is an intricate term for healthcare professionals. For example, 72.5% of healthcare professionals in the UK reported that they would like to have more knowledge of FASDs [33]. 

Not only the lack of knowledge about prenatal alcohol consumption but also stigma and reluctance to report alcohol use contribute to challenges in prevention and diagnosis. The data suggest that women who have a child with an FASD and individuals with FASDs are stigmatized [34]. The data from one study indicated that mothers of children with FASDs were considered more different, disdained, and responsible than women with serious mental illnesses, substance use disorders, or jail experiences [35]. Stigma may manifest as a public view that children with FASDs will inevitably be societal failures due to engagement in criminal behavior and the use of alcohol and drugs [34]. Public stigma has significant negative effects. For example, due to societal judgment, mothers of children with FASDs conceal having consumed alcohol during pregnancy, which leads to underdiagnosis and inadequate care [36]. Furthermore, negative public attitudes toward children with FASDs negatively impact their self-esteem, which may lead to isolation, dropping out of school, or turning to alcohol and drugs [37]. Research shows that due to stigma, children with FASDs often feel misunderstood, disrespected, bullied, and blamed [34]. Clinicians need to elaborate on acceptable ways to discuss with children and adolescents with FASDs about their difficulties. Talking to mothers about alcohol use during pregnancy reveals complicated topics that clinicians may want to avoid. Generally, it is a challenging theme for patients to discuss with doctors and doctors with patients, and this may vary according to the sociocultural context. An especially sensitive issue arises when clinicians think about how to speak to children and adolescents about their FASD features and how to communicate with their parents or guardians.

The literature review demonstrated that many strategies are undertaken to prevent FASD. The primary prevention goal is to eliminate the causes of FAS/FASDs by not exposing a fetus to alcohol. The data show that there is no safe amount of alcohol for a fetus [3,38]. All women should abstain from alcohol during pregnancy or before conception; therefore, all childbearing-age women should be educated about alcohol consumption. Family members and partners should also be educated on being supportive of a healthy pregnancy [6,39]. Good examples of prevention could be shared with other institutions and countries. Flyers, posters, media campaigns, events, and labels on alcoholic drink containers can reach the population’s awareness worldwide. 

Several prevention methods target women at risk. Two of the most effective primary prevention programs are the Changing High-Risk Alcohol Use and Increasing Contraception Effectiveness Study (CHOICES) by the Centers for Disease Control and Birth Control and Alcohol Awareness: Negotiating Choices Effectively (BALANCE) [40,41]. Both interventions work on reducing alcohol consumption and increasing the use of effective contraception among childbearing-age women at risk of alcohol-exposed pregnancies, which it defines as women who recently had vaginal intercourse and regularly consume higher quantities of alcohol. CHOICES and BALANCE use motivational interview sessions to change women’s perspectives and behavior and contraceptive counseling sessions. The World Health Organization (WHO) Regional Office for Europe conducted a rapid review of studies performed from 2005 to 2015, which targeted pregnant and nonpregnant women who are at risk of alcohol-exposed pregnancies. Eleven studies were performed in the USA and two in South Africa using the CHOICES and BALANCE methods. The studies targeted women who consumed alcohol in the last month and were at risk of being pregnant by not using effective contraception. The interventions, lasting from 20 to 60 min, concerning alcohol consumption and effective birth control were performed. Both interventions showed a significant reduction in the risk group [42].

Other programs may also be effective. For example, a team of mentors in the Parent–Child Assistance Program (PCAP) helped women overcome their addictions in a cycle of three years [43]. Another program, called the EARLY, divided women into three groups. One group received a face-to-face intervention of 60 min; the second group watched a video and received a 5 min intervention; and the third group received informational brochures. After a follow-up, the most significant decrease in risk was seen for those who received face-to-face counseling [44].

Alcohol consumption during pregnancy could come together with maternal polysubstance abuse, and these conditions bring many unanswered questions, both for clinicians and researchers. It is challenging for clinicians to talk to mothers about alcohol and drug use during pregnancy, and this can lead to severe consequences for the babies. Neonatal abstinence syndrome (NAS), or neonatal opioid withdrawal syndrome (NOWS), results from acute discontinuation of transplacental opioid exposure following delivery. Infants with NAS are at risk for long-term mental and physical health problems. Therefore, infants will benefit from connections with a primary care provider before hospital discharge and entities designed for early childhood intervention and developmental assistance [45]. Long-term interdisciplinary cooperation is needed to provide adequate comprehensive complex assistance.

It is necessary to apply the methods of treatment and assistance described in the literature, which have not yet been thoroughly investigated and identified, and the search for effective treatment methods is still ongoing. The existing data suggest that iron deficiency contributes to the severity of FASDs and provide a mechanistic explanation linking these two conditions [46]. The research conducted to date also highlights avenues for future investigation to further elucidate the mechanisms via which choline supplementation in the prenatal and postnatal periods may improve long-term neurodevelopmental outcomes [47]. Art therapies can also improve brain function and serve as a valuable method for planning complex care in FASDs. A pilot study examining the effects of music training on attention in children with FASDs showed promising results [48].

A promising case report described a good postoperative recovery of a newborn that underwent a surgical procedure with resection of the coarcting zone and anastomosis [49]. A fetal ultrasound examination was helpful to detect probable fetal alcohol syndrome, fetal abnormalities, and growth retardation. Despite the favorable cardiac and hemodynamic progression, the neurodevelopmental prognosis of this child remained uncertain because of fetal alcohol syndrome and exposure to several toxins [49]. In such complex cases, the child’s psychoneurological development often remains disturbed and requires long-term monitoring and treatment as well as psychosocial support.

The epidemiology of FASDs shows that the highest prevalence ratio is in the European Region [14]. Another study revealed the prevalence of FASDs among children adopted from Eastern European Countries, including Russia and Ukraine [50]. At least half of the children in that study group had an FASD diagnosis. Due to significant physical, mental, and functional damage caused by prenatal alcohol exposure, these children and their families might require longitudinal and complex treatment and educational and social support. A considerable prevalence in Poland and serious health consequences indicate the need to develop national FASD guidelines that align with international standards but are adjusted to the Polish context [51]. The authors state that “to make recommendations acceptable and valuable for national specialists, these must be elaborated by the interdisciplinary group of professionals representing various groups of stakeholders, taking into account the differences among validated clinical diagnostic systems for FASD in the world, the lack of single biochemical or imaging objective test for FASD, and the specific national healthcare context”. These statements could be relevant for the organization of services in many countries. Nevertheless, many questions remain concerning the concept of FASDs, especially in milder cases in older children with comorbidities. Even the most recent research suggests seeking a consensus on diagnostic criteria and evidence-based treatments and describing the pathophysiology and lifelong effects of FASDs [1].

Another critical and sensitive issue is the need for a broader elaboration of relevant and sensitive tactics for communicating and building therapeutic relationships with FASD patients and their family members. Due to these difficult topics, clinicians in Lithuania tend to avoid straight and open communication about the disorder with children and adolescents. Clinicians often find it difficult and uncomfortable to talk to patients, their parents, or guardians about FASDs, especially about the harm caused by alcohol consumption during pregnancy. Due to cultural contexts, stigma, and negative public opinion, diagnosis is avoided and missed, thinking that denying a diagnosis could be beneficial to the social adjustment of the patient. This is an obstacle to elaborating relevant treatment pathways and estimating the scope of the need for assistance. The key message to convey to patients with FASDs remains unclear and little researched. Whether patients with FASDs are more sensitive to the effects of alcohol and should avoid it in their lives or consume less than others has not been established. 

Because FASDs are associated with learning disabilities, it is necessary to raise awareness not only among medical professionals but also educators. Adopted children, and especially teenagers, need complex support, including in the medical, educational, and social sectors. Prevention programs for adopted children with FASDs need individual approaches to find appropriate sociocultural aspects and highlight sensitivity to alcohol. Clinical and research attention should be increased to overcome the stigma associated with FASDs and be focused on the necessary recognition and medical support at an early age and the psychosocial and educational aspects in adolescence. There is also a lack of research on the health and quality of life of young adults with FASDs.

## 5. Conclusions

Alcohol consumption during pregnancy and fetal alcohol spectrum disorders are common yet preventable public health issues. FASDs are highly multidisciplinary conditions that affect many aspects of a child’s physical, somatic, cognitive, emotional, and psychosocial development. Primary prevention strategies for FASDs should be implemented as soon as possible as they are effective. Prenatal and early postnatal diagnosis and individualized interdisciplinary treatment are essential but still underdeveloped. The literature highlights a lack of knowledge about FASDs and the negative consequences of prenatal alcohol exposure among the public, as well as among women of child-bearing age and healthcare professionals. Stigma, cultural contexts, knowledge gaps, and the heterogeneity of clinical manifestations are significant barriers to an accurate diagnostic process. The follow-up care of people with FASDs requires an individually planned, complex, and multidisciplinary approach. Raising awareness among clinicians on the developmental importance of the recognition of FASD symptoms may help overcome the stigmatized misconceptions that prevail in society about the complicated social adaptation after FASD diagnosis. The research notes that further investigations on this topic are needed.

## Figures and Tables

**Figure 1 children-11-00005-f001:**
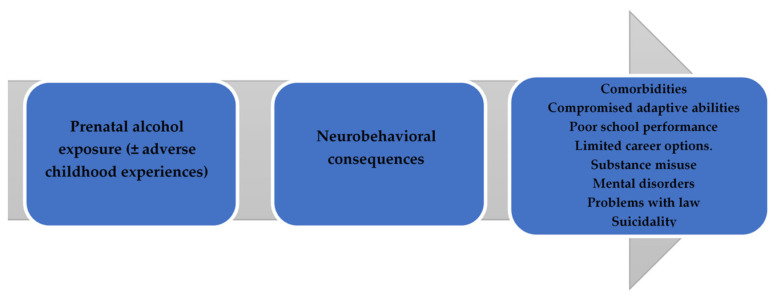
Proposed pathway of biopsychosocial consequences of FASDs. Alcohol has direct teratogenic effects on the fetal nervous system. Strengthened by a negative environment in childhood, the latter neurobehavioral consequences may lead to one’s maladaptive functioning (such as difficulties in communication, daily living skills, and socialization). Children with FASDs face difficulties at school and suffer from other diseases, such as ADHD. This leads to negative social consequences such as unemployment, alcohol and drug abuse, and problems with the law.

**Figure 2 children-11-00005-f002:**
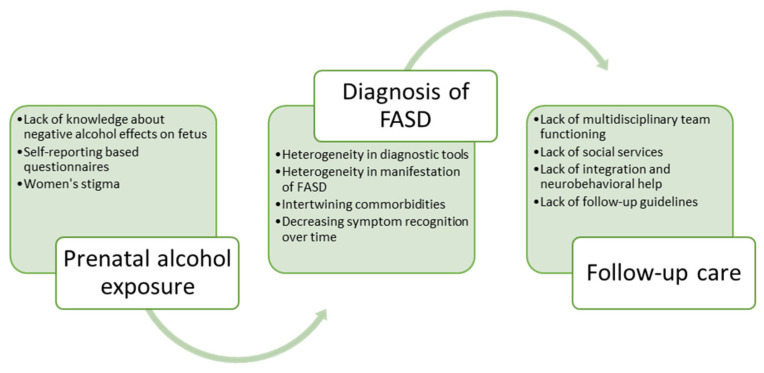
Difficulties in assessment of prenatal alcohol exposure (PAE), diagnosis of FASDs, and post-diagnosis care.

**Table 1 children-11-00005-t001:** Diagnostic criteria of FASD umbrella terms.

Diagnosis	Documented Prenatal Alcohol Exposure	Facial Dysmorphology	Growth Deficiency	Central Nervous System Dysfunction	Neurobehavioral Impairment
Fetal alcohol syndome	+/−	+	+	+	+
Partial fetal alcohol syndrome	+	+	+	−	+
Partial fetal alcohol syndrome	+	+	−	+	+
Partial fetal alcohol syndrome	+	+	−	−	+
Partial fetal alcohol syndrome	-	+	+	−	+
Partial fetal alcohol syndrome	-	+	-	+	+
Alcohol-related neurodevelopmental disorder	+	-	-	−	+
Neurobehavioral disorder associated with prenatal alcohol exposure	+	−	−	-	+
Alcohol-related birth defects *	+	−	−	+/−	+/−

* Must contain one or more congenital defects (malformations or hypoplasia of the bones, heart, kidney, hearing system, and vision. Sources: [10,12,13].

## Data Availability

There are no new data associated with this article.

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
