# Peer review of "Fetal Alcohol Spectrum Disorders and Inadequacy of Care: Importance of Raising Awareness in Clinical Practice"

_children, 2023, doi:10.3390/children11010005_

Round 1

Reviewer 1 Report

Comments and Suggestions for Authors

1# I would like to suggest the authors to check the instructions whether they can use ‘Prof.’ title before their names.

2# The authors name appeared twice (page 1, line 6)

3# In the instructions it is clearly written that the abstract should not contain any headings as mentioned bellow –

‘The abstract should be a single paragraph and should follow the style of structured abstracts, but without headings: 1) Background: Place the question addressed in a broad context and highlight the purpose of the study; 2) Methods: Describe briefly the main methods or treatments applied. Include any relevant preregistration numbers, and species and strains of any animals used; 3) Results: Summarize the article's main findings; and 4) Conclusion: Indicate the main conclusions or interpretations.’

The authors are suggested to recheck the instructions and rewrite to correct.

4# The ‘Introduction’ is too short in length. The authors need to think about it.

5# In the ‘Methods’ section the authors wrote ‘An extensive literature review has been performed. Only articles that have been published from 2000 to 2022 were included in the review.’ What is the rational to choose articles only for two years span? The authors need to share their thoughts with readers. Is any article published before with the same issue???

6# Font size of figure 1 is too small.

7# The following lines are presented in page no 5, line no 179-183. “Children with dysmorphic features of FASD can be identified as early as 9 months of age with a peak of identification of 18 months of age compared to the unaffected group of children. However, children with severe impairment of dysmorphic features can be identified at birth. While early behavioral and development impairment is usually noted between 18 and 42 months of age, CNS impairment may not be apparent until the child is in school [22].”

The authors cited only one article. Is the ref 22 representing all those sentences? If so, the authors are suggested to repeat ref 22 in earlier sentences as well.

8# Overall, the idea of this manuscript is generalized. It seems that everybody knows this fact. The catchy thing of this manuscript is the proposed ways to reduce the gaps between care and clinical practices. However, the ways they mentioned is also generalized. Another fact is the number of supportive articles is less than usual.

Author Response

The authors are very thankful to the reviewer for the valuable notes. We have made necessary changes accordingly and will reply briefly point by point:

1# I would like to suggest the authors to check the instructions whether they can use ‘Prof.’ title before their names.

1# We have deleted Prof. titles that were written in the line of the authors list. Sigita Lesinskiene and Algirdas Utkus are two authors who are professors at Vilnius University, title Prof. appeared by mistake, thanks for the noticing, of course, there should be only names and surnames of authors written, and we’ve corrected that place.

2# The authors name appeared twice (page 1, line 6)

2# Authors' names appear only once, we corrected that place.

3#In the instructions it is clearly written that the abstract should not contain any headings as mentioned bellow

3# The abstract is corrected accordingly, and headings have been removed.

4# The ‘Introduction’ is too short in length. The authors need to think about it.

4# The Introduction: the text is supplemented, expanded, and augmented with five new references.

5# In the ‘Methods’ section the authors wrote ‘An extensive literature review has been performed. Only articles that have been published from 2000 to 2022 were included in the review.’ What is the rational to choose articles only for two years span? The authors need to share their thoughts with readers. Is any article published before with the same issue???

5# Artickles published 2000-2023 were included, there is a  23-year span (not two), which is enough to see the developments in the field. It appeared that the amount of literature is not big, and many unanswered questions remain.

6# Font size of figure 1 is too small.

6# Figure 1 has been changed with a bigger font size.

7# The following lines are presented in page no 5, line no 179-183. “Children with dysmorphic features of FASD can be identified as early as 9 months of age with a peak of identification of 18 months of age compared to the unaffected group of children. However, children with severe impairment of dysmorphic features can be identified at birth. While early behavioral and development impairment is usually noted between 18 and 42 months of age, CNS impairment may not be apparent until the child is in school [22].”

7# Citation is corrected. 

8# Overall, the idea of this manuscript is generalized. It seems that everybody knows this fact. The catchy thing of this manuscript is the proposed ways to reduce the gaps between care and clinical practices. However, the ways they mentioned is also generalized. Another fact is the number of supportive articles is less than usual.

8# More ideas and specific descriptions are added in the Discussion and Conclusions sections. A literature search has shown that there is not much literature on FASD, and further research is needed. Our manuscript aimed to provide a narrative approach to the literature, tackling many aspects in one comprised place that usually are described separately.

On behalf of all authors, sincerely,

Reviewer 2 Report

Comments and Suggestions for Authors

The paper is presenting a narrative approach to health research on FASD. I am not familiar with this approach. However, the paper is presenting comprehensive information on aspects of FASD such as diagnostic problems (FASD letter soup), and suggestions to improve diagnosis and care.

Just one minor suggestion: Traditionally, the Discussion part is to discuss Results. It does not present new oder different results. Please shorten the Discussion by relocating the studies mentioned for the first time in Discussion to Results.

Comments on the Quality of English Language

I think some language editing is helpful.

Author Response

The authors are thankful to the reviewer for the valuable notes. We have made the necessary changes. Accordingly, we will reply here briefly to each note point by point:

The paper is presenting a narrative approach to health research on FASD. I am not familiar with this approach. However, the paper is presenting comprehensive information on aspects of FASD such as diagnostic problems (FASD letter soup), and suggestions to improve diagnosis and care.

 Just one minor suggestion: Traditionally, the Discussion part is to discuss Results. It does not present new oder different results. Please shorten the Discussion by relocating the studies mentioned for the first time in Discussion to Results.

We considered your observations and added 5 more new sources to the literature covering the missing aspects you noted. We rewrote and edited the discussion section considerably. In the Discussion section, we have left some new literature citations that draw on scholarly sources that illustrate that particular aspect when we develop the Discussion of FASD.

I think some language editing is helpful.

According to your recommendation, the manuscript was edited by MDPI‘s English editing service.

On behalf of all authors, sincerely,

Reviewer 3 Report

Comments and Suggestions for Authors

Congratulations to the authors. Excellent work!

Minor comments:

The abstract should follow the style of structured abstracts, but without headings, please revise.

Indicate whether your literature review was systematic or non-systematic in the methodology section. If your review is systematic, please include the PRISMA methodology statement. Additionally, inclusion and exclusion criteria for the utilized articles would be beneficial.

Line 39: FASD abbreviation-abbreviated text wasn’t mentioned before the abbr. Please provide an additional explanation of the abbreviation.

Line 165: Same. Please provide an explanation of the abbreviations CAGE, TWEAK, AUDIT-C, T-ACE, and T-ACER3. Moreover, omit abbreviations since they are no longer mentioned in the text.

Author Response

The authors are thankful to the reviewer for the valuable notes. We have made the necessary changes. Accordingly, we will reply here briefly to each note point by point:

The abstract should follow the style of structured abstracts, but without headings, please revise.

The abstract was revised and now is without the headings.

Indicate whether your literature review was systematic or non-systematic in the methodology section. If your review is systematic, please include the PRISMA methodology statement. Additionally, inclusion and exclusion criteria for the utilized articles would be beneficial.

Our literature is a non-systematic narrative review. A literature search has shown that there is not much literature on FASD, and further research is needed. Our manuscript aimed to provide a narrative approach to the literature, tackling many aspects in one comprised place that usually are described separately.Only peer-reviewed publications written in English from 2000 to 2023 were included in the review.

Line 39: FASD abbreviation-abbreviated text wasn’t mentioned before the abbr. Please provide an additional explanation of the abbreviation.

This was corrected.

Line 165: Same. Please provide an explanation of the abbreviations CAGE, TWEAK, AUDIT-C, T-ACE, and T-ACER3. Moreover, omit abbreviations since they are no longer mentioned in the text.

These corrections were done, and the text explained the abbreviations. The abbreviations added to the fill title help to show them because they are often used in the FASD literature.

MDPI‘s English editing service edited the manuscript.

On behalf of all authors, sincerely,

Round 2

Reviewer 1 Report

Comments and Suggestions for Authors

It looks odd to add the 'Result' section in a narrative review. Authors are suggested to replace it with any other broad heading.

Author Response

Many thanks for the note. We have changed the title of the section: Results we have changed into Recent Findings.